# How Elongator Acetylates tRNA Bases

**DOI:** 10.3390/ijms21218209

**Published:** 2020-11-03

**Authors:** Nour-el-Hana Abbassi, Anna Biela, Sebastian Glatt, Ting-Yu Lin

**Affiliations:** 1Malopolska Centre of Biotechnology, Jagiellonian University, 30-387 Kraków, Poland; norhane.abbassi@gmail.com (N.-e.-H.A.); anna.biela.banas@gmail.com (A.B.); 2Postgraduate School of Molecular Medicine, Medical University of Warsaw, 02-091 Warsaw, Poland

**Keywords:** Elongator, Elp3, tRNA modification, acetyl-CoA, proteome balance, cancers, neurodegenerative diseases

## Abstract

Elp3, the catalytic subunit of the eukaryotic Elongator complex, is a lysine acetyltransferase that acetylates the C5 position of wobble-base uridines (U_34_) in transfer RNAs (tRNAs). This Elongator-dependent RNA acetylation of anticodon bases affects the ribosomal translation elongation rates and directly links acetyl-CoA metabolism to both protein synthesis rates and the proteome integrity. Of note, several human diseases, including various cancers and neurodegenerative disorders, correlate with the dysregulation of Elongator’s tRNA modification activity. In this review, we focus on recent findings regarding the structure of Elp3 and the role of acetyl-CoA during its unique modification reaction.

## 1. Introduction

Acetyl coenzyme A (acetyl-CoA) is an essential cofactor that serves as an acetyl group donor and is crucial for cellular metabolism and specific regulatory cascades. For instance, protein acetylation regulates enzymatic activities and guards the mitochondrial TCA cycle [1]. The enzymes responsible for catalyzing the underlying acetylation reaction belong to the acetyltransferase protein family. Two distinct sites of acetylation are known in proteins—namely, (i) the α-amino group at the *N*-terminus and (ii) the ε-amino group at the side chain of lysine residues. The former is catalyzed by *N*-terminal acetyltransferases (NATs) and the latter is executed by lysine acetyltransferases (KATs). NATs belong to the GCN5-related *N*-acetyltransferases (GNAT) superfamily, which in fact also includes some KATs. NATs, such as NatA-NatF, most often exist as multimeric macromolecular complexes. For instance, NatA is composed of *N*-alpha-acetyltransferase 10 (Naa10) and the auxiliary subunit Naa15, whereas the human NatE complex consists of NatA, Naa50, and the Huntingtin-interacting protein K [2]. The acetylation of protein *N*-termini occurs co-translationally on the emerging nascent peptides and represents an irreversible process on 70–90% of all human proteins. This modification was shown to affect various properties of the target protein, including the half-life time, intracellular localization, or affinity to interaction partners [3].

Lysine acetylation, one of several post-translation modifications (PTM) affecting naturally occurring amino acids in proteins, describes the attachment of an acetyl group from acetyl-CoA to the ε-amino group of a lysine side chain. Due to the increased length of the side chain, the reduced positive charge, and the altered electrostatic properties of the protein, the decoration with an acetyl group can directly change the activities of target enzymes by influencing protein–protein or protein–nucleic acid interactions. For instance, core histones lose their ability to bind DNA when their tails are acetylated, facilitating the active gene transcription of the affected genomic regions. The acetylation of α-tubulin (at lysine 40) is yet another well-characterized example which prolongs the half-life time of tubulin and allows the stabilization of the lattice structure of microtubules [4]. In total, we currently know of more than 2000 lysine acetylated proteins in the literature, including central hub proteins such as p53, c-MYC, and NF-κB. Hence, protein acetylation appears to be one of the key mechanisms to guide, control, and regulate a plethora of cellular activities and functions post-translationally [5].

There are three classical KAT families, formerly called histone acetyltransferases (HATs)—namely, GNAT, MYST, and p300/CBP. The catalytic domains of all KATs share a similar tertiary α/β-fold structure. The core α/β-fold is essential for acetyl-CoA binding, substrate recognition, and the subsequent transfer of the acetyl group during the modification reaction. Although KATs share a similar core structure, their downstream targets as well as their catalytic mechanisms are extremely diverse. GNAT and MYST enzymes utilize a sequential mechanism that employs a conserved active-site glutamate/aspartate residue to deprotonate the *N*-ε-lysine of histone tails, which facilitates a nucleophilic attack on the bound acetyl-CoA [6]. p300/CBP enzymes contain an acidic substrate binding surface that can accommodate different sets of substrates, and a tyrosine residue has been proposed to function as a general acid during catalysis [7]. The state of lysine acetylation is reversible, and the modification can be removed by enzymes called lysine deacetylases (KDACs). The interplay between KATs and KDACs creates a complex but well-balanced equilibrium that controls a large number of cellular processes [3].

Apart from its key role during protein acetylation, acetyl-CoA is also required to modify RNA bases in messenger RNAs (mRNAs), ribosomal RNAs (rRNAs), and tRNAs. N^4^-acetylcytidine (ac^4^C) can be found in 18S rRNA, tRNAs, and mRNAs [8], whereas 5′-carboxymethyluridine (cm^5^U) is detected at uridines in the wobble position of tRNA (U_34_). The cm^5^U_34_ modification is the precursor of several derivatives, including 5-methoxycarbonylmethyl-uridine (mcm^5^U_34_) and 5-carbamoylmethyl-uridine (ncm^5^U_34_). In a subset of mammalian tRNAs, the mcm^5^U_34_-modified wobble base is further processed to 5-methoxy-carbonyl-methyl-2-thio-uridine (mcm^5^s^2^U_34_) or 5-methoxy-carbonyl-hydroxymethyl-uridine (mchm^5^U_34_). These chemical modifications provide additional spatial restraints for the codon-anticodon base pairing during ribosomal decoding and translation elongation. Therefore, they contribute to the kinetics of the ribosomal elongation cycle and affect the co-translational folding dynamics and protein folding. In contrast to the reversible nature of acetylation events on proteins, the existence of de-modification enzymes that can remove ac^4^C or cm^5^U groups from the RNA bases has not yet been reported.

The generation of ac^4^C is catalyzed by NAT10 [8], whereas the cm^5^U_34_ modification reaction is conducted by the Elongator complex [9]. As the catalytic subunit of Elongator (Elp3) shows a high sequence similarity to Gcn5, it was long believed that Elongator acetylates proteins. Only recently was it convincingly shown that it indeed acts as a non-canonical acetyltransferase that catalyzes reactions on tRNAs instead of proteins [10,11,12,13]. In contrast to the formation of a carbamoyl group by common protein acetyltransferases, Elp3 connects the methyl group of the acetyl residue with the C5 atom of the targeted uracil base. This type of reaction is unique to Elp3 proteins and, in addition to a KAT domain, it also requires an additional radical S-adenosyl methionine (rSAM) domain to execute its full modification activity. As the priming cm^5^U_34_ modification by Elongator is the first step in some xcm^5^U_34_ derivatives formation, ablation of Elongator causes loss of all xcm^5^U_34_ derivatives. The lack of these modifications leads to proteome aggregation, proteotoxic stress and the mis-regulation of various key proteins involved in a large variety of cellular functions, including the initially suspected transcription regulation [14]. Furthermore, a reduction in Elongator-dependent tRNA modification has been linked to the onset of severe human pathologies, including cancers and neurological disorders [15]. Here, we aim to provide a comprehensive summary of the recent developments around the highly conserved catalytic Elp3 subunit from bacteria, archaea, and eukaryotes. We focus on the available structural and biochemical details and link these new data to existing clinical evidence. The implications of Elongator dysfunction in various organisms and the pathological consequences of other tRNA modification enzymes are well-summarized by recently published reviews [16,17,18].

## 2. Elp3 Is the Catalytic Core of Elongator

### 2.1. Elp3 Exists in Three Domains of Life

Elongator is a macromolecular complex that consists of two copies of each of its six subunits (Elp1-6). Elp3 (EC 2.3.1.48), also named KAT9, is recognized as the catalytic core of the complex, harboring the KAT domain (Figure 1) [19]. Historically, several studies have linked Elongator also with the acetylation of histones [20], α-tubulin [21], and other proteins [22,23]. In contrast to the role of Elongator as a protein acetyltransferase, an early study surprisingly linked Elongator to the cm^5^U_34_ modification of tRNAs [9]. In recent years, accumulating evidence showed that a plethora of cellular and pathological phenotypes all indirectly result from its lack of genuine U_34_ modification activity [24].

The eukaryotic Elongator complex is a large protein machine which is highly conserved among Amoebozoa, plants, fungi, animals, and humans [17]. Elp1, the largest subunit, exists as a dimer and its *N*-terminus serves as a scaffold for Elp2 and Elp3. Elp1 also interacts with Elp4, and this interaction seems to be involved in the dynamic association and dissociation of the Elp456 subcomplex during catalysis [25]. The C-terminal domain (CTD) of Elp1 contains a conserved basic region which acts as a tRNA binding element [26], but the details of tRNA selectivity and rejection are not yet clarified [10]. Elp2 consists of two WD40 domains, and its rigid structure and integrity seem to confer additional stability to the other subunits [25]. Elp456 forms a hexameric ring and it has intrinsic ATPase activity, which enables it to regulate tRNA binding independently of Elp123 [27]. The dissociation of tRNA from Elp456 is coupled to its ATP hydrolysis activity, which indicates a dynamic exchange of substrate tRNAs between the two subcomplexes during the reaction cycle. A crosslinking-coupled mass spectrometry approach has confirmed that the transient interaction between Elp123 and Elp456 is mainly mediated via Elp1/Elp4 and Elp3/Elp4 [25]. Moreover, the negative stain electron microscopy reconstructions of the fully assembled complex revealed that Elp456 is asymmetrically positioned on one of the Elp123 lobes and sits on top of the active site of Elp3 [25,28]. This supports the proposed model of Elp456 acting as a tRNA recruiter before or a tRNA remover after the modification reaction.

Archaea also carry Elp3 genes in their genomes [11,13], but other Elongator subunits are absent in these organisms. Since the existence of mcm^5^U_34_ in tRNAs has been documented in the archaeon *Haloferax volcanii* [29], it has been suggested that Elp3 by itself is capable of catalyzing the first step of the tRNA modification reaction. In addition, the cm^5^U_34_ modification reaction has been recapitulated in vitro using purified *Methanocaldococcus infernus* Elp3 (*Min*Elp3) [13]. Most bacteria employ the MnmE/MnmG complex to catalyze different types of xm^5^U_34_ modifications in tRNAs—namely, 5-carboxymethylaminomethyluridine (cmnm^5^U; using glycine) or 5-aminomethyluridine (nm^5^U; using ammonium). Strikingly, Elp3 is also found in one specific anaerobic bacterial clade (*Dehalococcoides* spp) inhabiting toxic waste areas. Since cultivating these bacteria in laboratory conditions is difficult, we lack direct evidence of the existence of xcm^5^U_34_ derivatives in these bacterial species [11].

### 2.2. Elp3 Protein from All Domains of Life Have the Same Structure

Apart from bacterial, archaeal, and yeast Elp3s, the enzymatic activities of Elongator have been investigated in many other eukaryotic organisms, including worms, flies, zebrafish, plants, mice, and humans [30]. However, quantitative biochemical and biophysical in vitro analyses of Elongator’s modification activities have been limited by severe technical challenges. Neither recombinantly expressed systems nor the reconstitution of the complex with individually purified components or approaches involving the purification of endogenous complexes from large-scale yeast fermentation have yielded sufficient quantities of purified eukaryotic Elongator complexes for comprehensive analyses. In contrast to the multi-subunit eukaryotic Elongator complex, individual prokaryotic and archaeal Elp3 proteins, such as *Dehalococcoides mccartyi* Elp3 (*Dmc*Elp3), *Methanocaldococcus jannaschii* Elp3 (*Mj*Elp3), and *Min*Elp3, can be produced in large quantities in bacteria and can be purified [11,12,13,31]. The availability of these autonomous Elp3 proteins has strongly furthered our understanding of Elp3′s function, atomic structure, and role in the reaction cycle of the fully assembled Elongator complex in eukaryotes.

In detail, Elp3 possesses two functional domains—namely, a radical S-adenosyl methionine (rSAM) binding domain and a KAT domain. In recent years, the atomic structures of *Dmc*Elp3, *Min*Elp3, and *Saccharomyces cerevisiae* Elp3 (*Sc*Elp3) were determined using crystallography or single-particle cryo-EM analysis [10,11,12]. These proteins not only share a high sequence similarity but also seamlessly adopt the same spatial domain arrangement in their nearly identical structures (Figure 1). In detail, the rSAM and KAT domains are held together by an extensive hydrogen bond network, forming a cleft between them to accommodate the tRNA substrate. The rSAM domain, which is homologous to the bacterial RlmN methyltransferase, coordinates an iron-sulfur cluster via three strictly conserved cysteine residues [11,31]. The cryo-EM snapshot of yeast Elp3, embedded in the Elp123 subcomplex, also confirmed the identity of the iron sulfur cluster and its position within Elp3 proteins. Moreover, the structure contains density for a bound 5′-deoxyadenosyl molecule (5′-dA), the cleavage product of S-adenosylmethionine (SAM), located in the rSAM domain [10]. The structure of the KAT domain of Elp3 resembles other acetyltransferases that all share a similar protein fold, despite variations in their primary amino acid sequences. The Elp3 KAT domain is most similar to Gcn5 proteins, although it lacks two helices (α1 and α2) and harbors an additional seventh β-strand [11]. In addition, an “acetyl-CoA blocking loop” that occupies the canonical acetyl-CoA binding site is clearly visible in the *Dmc*Elp3 crystal structure. Similar sequence stretches are present in all Elp3 proteins, but the loops appear disordered in the *Min*Elp3 and *Sc*Elp3 structures. The unstructured N-termini of Elp3 proteins are the least conserved regions among Elp3 proteins, and their main function is still unclear. Despite the low conservation in the *N*-termini of Elp3s, there are multiple basic residues residing in this stretch, and regional truncations of the *Min*Elp3 *N*-terminus have shown decreased tRNA binding. Replacing the *N*-terminus of *Min*Elp3 with that of *Sc*Elp3 or *Hs*Elp3 restores tRNA binding to a certain extent. These data show that common features in these *N*-termini are responsible for tRNA binding and that a high sequence similarity might not be required to fulfill a similar function. Of note, the *N*-terminus is very short in *Dmc*Elp3, which may explain its lower tRNA binding affinity compared to that of other Elp3s [10,11]. As the “acetyl-CoA blocking loop” in *Dmc*Elp3 is in close contact with the residues of the *N*-terminus, these links could represent a regulatory switch between tRNA recruitment and acetyl-CoA binding in all Elp3 proteins [12]. Despite the high overall conservation of the different Elp3 structures, *Dmc*Elp3 is the only example that coordinates a zinc (Zn) ion. The equivalent region is either replaced with a disulfide bond between two cysteines in the loop of *Min*Elp3 or a helical element that completely lacks the responsible cysteine residues in *Sc*Elp3. This region, named “central linker” in *Min*Elp3, bridges the rSAM and KAT domains, significantly increases the domain interface, and maintains the structural rigidity and integrity in all Elp3 proteins.

### 2.3. tRNA Triggers Elp3-Mediated Acetyl-CoA Hydrolysis

The acetyl-CoA binding pocket resides in the KAT domain of Elp3, but its binding affinity towards acetyl-CoA is almost twenty times lower than is the case for members of the Gcn5 superfamily. This relatively low affinity (e.g., *Min*Elp3; K_d_ ~135 µM) may be caused by the previously mentioned acetyl-CoA blocking loop that competes with acetyl-CoA binding [12]. Although the topology of the binding pocket resembles that of Gcn5 proteins, the critical residues for co-factor binding as well as the catalytic reaction are not well conserved. A co-crystal structure of *Dmc*Elp3 with desulfo-CoA (DCA), an acetyl-CoA analogue, has provided insights into the binding of acetyl-CoA to Elp3 (Figure 2) [12]. The ligand-bound structure is well aligned to the complex formed between CoA and *Tetrahymena thermophila* Gcn5 (*Tt*Gcn5), and in both structures the pantothenic acid and β-mercaptoethylamine of the ligand point into the predicted catalytic center. The Elp3-DCA structure allowed the identification of several conserved residues in the active site that are in direct contact with the ligand (e.g., Lys77, Lys193, Glu386, Gly408, Gly410, and Tyr441) [12]. Among them, Gly408 and Gly410 are in close proximity to one phosphate group of DCA. It has further been shown that the equivalent residues in *Min*Elp3, including Lys266, Gln461, and Tyr517, are involved in ligand binding and the equivalent single amino acid substitutions in yeast lead to Elongator loss-of-function phenotypes [11].

The catalysis of acetyl-CoA hydrolysis by KAT proteins has been extensively studied [5,32]. Different KAT proteins conduct different catalytic reactions, and they have diverse substrate specificities. Despite their structural similarity and the utilization of the identical acetyl donor (acetyl-CoA), the KAT domains of Elp3 and Gcn5 employ a remarkably different chemical reaction to acetylate their substrates (Figure 2). Foremost, the KAT domain of Elp3 is not sufficient for the modification reaction, and the activity depends on the concerted action of both Elp3 domains—namely, KAT and rSAM. Of note, the substrate tRNA is bound by residues in the rSAM domain and not directly in the KAT domain. Independently of tRNA binding, Elp3 also generates a radical (5′-dA•) in the rSAM domain. In parallel, acetyl-CoA is hydrolyzed in the KAT domain upon tRNA binding. The 5′-dA radical reacts with the methyl group from acetyl-CoA and abstracts a hydrogen atom from its methyl group to generate acetyl radical (acetyl•). The formed acetyl radical is then added to the C5 of U_34_ via a C-C bond [13]. In this proposed model, the existence of specific intermediates, either tRNA-acetyl-Elp3 or acetyl-CoA-Elp3, still needs clarification and the general base remains to be identified. Another yet-unresolved issue in the catalytic reaction concerns the mechanism of how tRNA binding triggers acetyl-CoA hydrolysis in the KAT domain. During canonical Gcn5-mediated protein acetylation reactions, the lysine residue of the target peptide that is to be modified triggers acetyl-CoA hydrolysis [33]. Elp3, which catalyzes the acetylation of U_34_, is stimulated specifically by U_34_-containing substrate tRNAs, while tRNAs lacking a target uridine in position 34 consistently fail to induce acetyl-CoA hydrolysis in Elp3. However, the position of the U_34_ base in the Elp123-tRNA cryo-EM structure is close to the 5′-dA molecule bound in the rSAM domain and therefore relatively distant from the KAT active site. Hence, it is unlikely that the U_34_ base directly triggers the acetyl-CoA hydrolysis reaction, like the target lysine in peptides modified by other KATs. Interestingly, a conserved lysine residue of the rSAM domain (e.g., Lys150 in *Min*Elp3) was found to point towards the active site of the KAT domain. A biochemical analysis confirmed that the MinElp3_K150A_ mutant lacks acetyl-CoA hydrolysis activity, yet still retains its tRNA and acetyl-CoA binding abilities [12]. Hence, Elp3 seems to utilize this integrated trigger to mimic the reactive lysine residue of target peptides and initiate the hydrolysis reaction.

On the basis of an enzymatic assay that allows the monitoring of the tRNA-induced acetyl-CoA hydrolysis activity of Elp3, a comprehensive analysis was performed on its substrate selectivity and specificity [12]. Foremost, the hydrolysis activity was exclusively stimulated only by tRNAs. Peptides or other nucleic acid substrates (e.g., DNA) failed to activate Elp3. Although Elp3 can bind to a broad range of tRNA substrates, its hydrolysis activity is only triggered by U_34_-containing tRNAs, suggesting a substrate-specific effect on the integrated lysine “trigger” residue. This mechanism of activation has been confirmed in archaeal Elp3 and the yeast Elp123 complex [10,12]. As mentioned above, *Dmc*Elp3 harbors a significantly shorter *N*-terminus, reduced tRNA binding affinity, and a more coordinated “acetyl-CoA blocking loop”. These differences might have caused the very weak activity found in the tested bacterial Elp3 protein.

## 3. Elongator Dysfunction Is Linked to Disease

### 3.1. Defects in Elongator Disturbs Proteome Balance and Is Associated with Neurodegenerative Diseases

Impaired Elongator activity causes the codon-dependent mis-production of individual proteins and proteome imbalance, which is detrimental for many fundamental cellular functions [15]. Furthermore, diminished Elp3 expression levels are the cause of spine malformation in zebrafish [34] and Elp3 mutations were identified in a mutagenesis screen for the key players guiding axon formation, synaptic transmission, and neuronal survival in Drosophila development (Figure 3) [35]. Moreover, it was found that amyotrophic lateral sclerosis (ALS) patients express lower levels of Elp3 in their motor cortex, which is accounted for in proteome impairment. As a result, it causes the loss of several critical proteins, such as SOD1, TDP43, and FUS, that contribute to the axonopathy of ALS. The genomic deletion of Elp3 causes multiple phenotypes in different organisms, ranging from malformation in zebrafish to early embryonic lethality in mouse models and the lethality of *Drosophila* larvae at the pupal stage [36]. Nonetheless, the availability of conditional Elp3 knockout mouse lines has provided insight into the role of Elongator in the development of different neuronal subtypes and ear cells [22,37,38,39,40,41,42].

A recent study combined transcriptome and proteome analyses to propose a direct link between codon bias and codon usage that explains the effect of Elongator activity on specific target proteins [43]. The list of affected gene products includes Brca2, which is responsible for the homology-directed repair (HDR) of DNA double-strand breaks and Rif1, which is required for non-homologous end-joining (NHEJ). Moreover, mutations in the human SETX gene are associated with diverse neurodegenerative disorders and the elevated DNA damage that is found in ALS. Foremost, this study provides an initial model of how Elongator (mis)regulation is directly related to ER stress or unfolded protein response (UPR), that are often associated with neurological disorders. For instance, ER stress and UPR are the hallmarks of familial dysautonomia (FD) in mouse models. This is a disorder of the autonomic nervous system, which is correlated with Elp1 mutations [44]. Clinically relevant allelic variations of other subunits are also reported to be associated with several neuron-related disorders, including intellectual disability and autism spectrum disorder (Elp4) [45,46]. Furthermore, a mutation in mouse Elp6 (Elp6_L126Q_) was recently shown to reduce the stability of the whole Elp456 hexamer, which causes specific Purkinje neuron degeneration and an ataxic-like phenotype [47]. It has also been reported that Elp4 might contribute to Rolandic epilepsy, although another study could not find a correlation between Elp4 gene and centrotemporal spikes [48,49].

### 3.2. Elongator in Cancers

Elongator-dependent translational control seems to be a key element for tumor formation and tumor growth. A Wnt-dependent up-regulation of Elp3 has been detected in intestinal epithelia cells, where it promotes the expression of the Sox9 protein, which is critical to maintain the cancer stem cell subpopulation [39]. Recently, another study reported that the oncogenic expression of polyomavirus middle T protein increases the expression levels of Elp3 and cytosolic thiouridylases subunits 1/2 (Ctu1/2), enzymes responsible for U_34_ thiolation. The elevated levels of Elp3 and Ctu1/2 promote the translation of DEK oncoprotein and, as a result, it up-regulates the lymphoid enhancer-binding factor 1 transcription factor-dependent pro-invasive transcriptomic signature in breast tumors [50]. Similarly, the development of BRAF^V600E^-expressing melanoma cells in zebrafish also relies on the elevated expression of Elp3 [41]. The same study further used ribosome profiling to reveal that Elp3 influences the decoding of specific codons in the *HIF1A* mRNA. The Elp3-dependent translation of HIF1A protein thus contributes to invasive features and drug resistance in malignant melanoma. Additionally, a recent study revealed that patients that lost Elp1 through a germline mutation in combination with the constitutive activation of Sonic Hedgehog signaling are susceptible to medulloblastoma, a malignant brain tumor in children [51]. The authors have also confirmed that the tumors from these patients indeed exhibited Elongator-related phenotypes, such as reduced levels of U_34_ modifications, codon-dependent translational reprogramming, and the induction of the unfolded protein response. In addition, gallbladder cancer patients with a lower Elp5 expression exhibit poor response to a treatment with gemcitabine. Mechanistically, gemcitabine induces apoptosis via elevating p53 expression, which is driven by heterogeneous nuclear ribonucleoprotein Q (hnRNPQ). The loss of Elongator directly abolishes the Elongator-dependent translation of hnRNPQ/p53 and leads to a loss of drug sensitivity in gallbladder cancer [52]. To date, over a hundred of tumor-related sequence variations in the human Elp3 gene have been described and summarized in cancer genome databases (“The Cancer Genome Atlas” (TCGA, https://www.cancer.gov/), ICGC (https://dcc.icgc.org/) and COSMIC (https://cancer.sanger.ac.uk/cosmic) [53]). Patient-derived mutations in Elp3 that concern conserved residues were tested in yeast and found to affect the tRNA modification activity of Elongator. For instance, R404T (liver cancer), R424K (lung cancer), and D443N (esophageal cancer) were confirmed to compromise Elongator activity [11]. These initial experiments suggest that the mutations found in patients might have altered Elongator activity and predisposed the patients for the disease. Nonetheless, further functional studies are needed to clarify the precise contributions of these mutations to the specific diseases progression and to establish genomic mutations in Elongator subunits as reliable disease markers in the future.

### 3.3. Other Mechanisms Regulating Elongator Activity

PTMs on Elongator subunits have recently been discovered to take part in dynamically regulating its cellular activities. For instance, the CTD of Elp1 contains critical phosphorylation sites that seem to regulate the recruitment of substrate tRNAs [54] and accessory proteins, like Kti12 [55]. In melanoma, it has been demonstrated that the PI3K-mTORC2 pathway regulates Elongator activity via the phosphorylation of human Elp1 (Ser1174) [41]. Another recent study showed that human Elp3 can be phosphorylated on a tyrosine residue (Tyr202) located in the rSAM domain and that the presence of pTyr202 promotes cell growth [56]. In addition, several PTM on Elp3, including methylation on human Elp3 (Lys229) or the phosphorylation of mouse Elp3 (Ser161), were identified using proteomics. However, the specific cascades responsible for these modifications and their impact on Elongator’s activity still need future studies (Figure 3). Additionally, Elp4 harbors several Ser/Thr phosphorylation sites, which appear to be crucial for balancing the nutrient-dependent switch between TORC1 and TORC2 [57]. The affected residue (Ser114) along with its flanking region resides in a motif that is shown to contact Elp1 and Elp3. Additional biochemical studies will be needed to provide mechanistic details of the effect of Ser114 phosphorylation on the interplay between Elp123 and Elp456. This review only highlights the most recent studies and additional dedicated expert summaries will be required to provide an overview of the dynamic cascades of PTMs that regulate the activity of Elongator. Overall, we need to achieve a deeper insight into these mechanisms to understand the cellular communication between Elongator and other signaling cascades and to assess the effects of Elongator-based intervention therapies for human diseases.

## 4. Conclusions and Future Aspects

Acetyl-CoA is a central metabolite in the metabolism of nearly every living cell. The ratio of acetyl-CoA and CoA can determine the fate of cells. For instance, elevated acetyl-CoA levels favor histone acetylation and activate gene expression, whereas low acetyl-CoA levels arrest anabolic reactions. The discovery of acetyl-CoA-based modifications of the tRNA anticodon adds yet another key cellular function to the list. The elongator-mediated acetylation of uridines in the wobble-base position represents a direct link between acetyl-CoA metabolism and the regulation of ribosomal translation elongation, co-translational folding dynamics and the integrity of our proteomes. Despite the recently surfacing key role of Elongator in cancer progression and metastasis, diagnostic tools and pharmaceutical agents that capitalize on this link have yet to be developed.

## Figures and Tables

**Figure 1 ijms-21-08209-f001:**
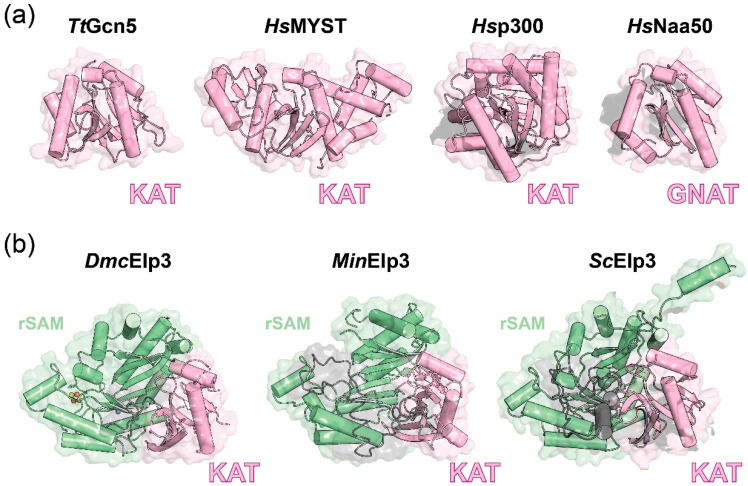
Protein acetyltransferases and Elp3s structures. (**a**) Structures of KAT domain from TtGcn5 (PDB: 1QSN), HsMYST (PDB: 2PQ8), and Hsp300 (PDB: 5LKT) and the GNAT domain from HsNaa50 (PDB: 6PPL). (**b**) Elp3 structures from *D. mccartyi* (PDB: 5L7J), *M. infernus* (PDB: 6IAD), and *S. cerevisiae* (PDB: 6QK7). Domains of each enzyme are as indicated.

**Figure 2 ijms-21-08209-f002:**
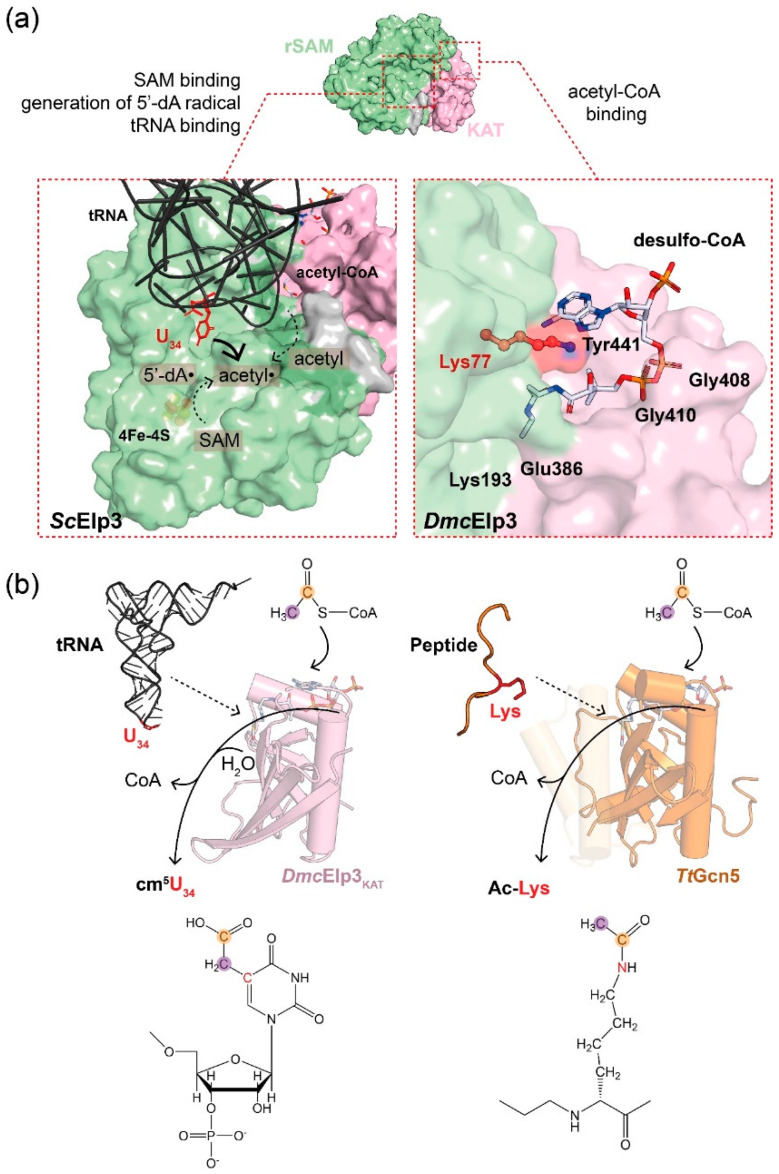
Biochemical features of Elp3. (**a**) Illustrations of the Elp3-dependent cm^5^U modification reaction (left) and the crystal structure of the acetyl-CoA binding pocket of DmcElp3 with desulfo-CoA (right, PDB: 6IA6). The components in the cm^5^U reaction are as indicated. The ligand contacting residues are labeled, while the conserved lysine (Lys77 in DmcElp3) is highlighted in red. (**b**) Schemes of the proposed tRNA-triggered acetyl-CoA hydrolysis of Elp3 (left) and Gcn5-mediated protein acetylation mechanism (right). The modifiable U_34_ and reactive K_14_ of histone H3 peptide are colored in red. For a clearer presentation, the α1 and α2 helices of TtGcn5 are shown in a transparent style. Acetyl-CoA, shown in cartoon, is modeled in the binding pockets of Elp3 or Gcn5, whereas the chemical structure of acetyl-CoA is shown on the top and the reactive carbons are highlighted in an orange circle (acetyl radical for cm^5^ modification) or purple circle (carbonyl group of the thioester of acetyl-CoA). The reactive sites of cm^5^U_34_ and Ac-Lys are both colored in red and present the C-C bond for the cm^5^ addition and the C-N bond for the acetyl-lysine addition, respectively.

**Figure 3 ijms-21-08209-f003:**
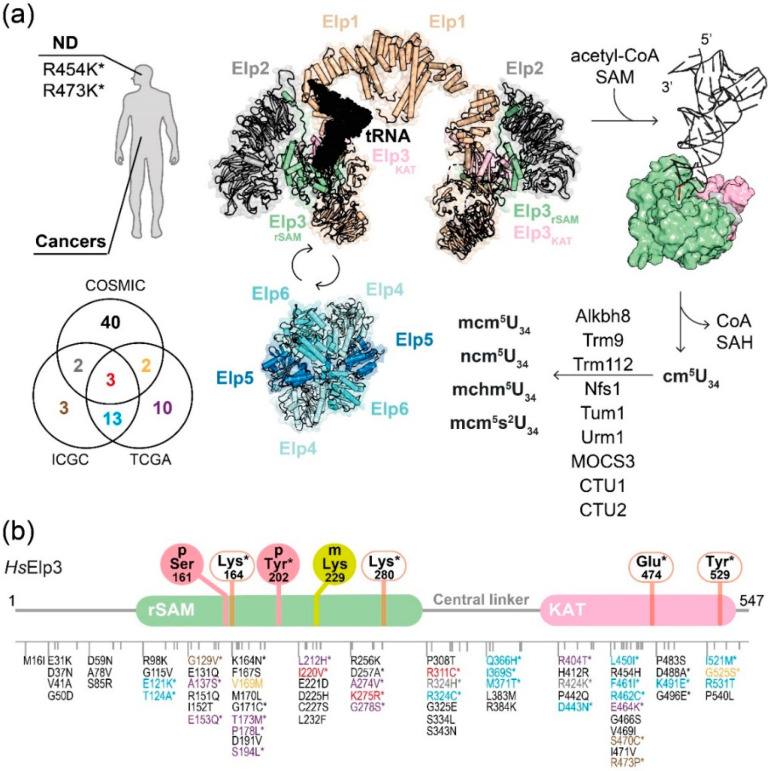
Elongator-mediated tRNA modification and clinically relevant mutations of human Elp3. (**a**) Schemes of disease-related mutations and Elongator-mediated tRNA modification cascade. The reported changes in human Elp3 from neurodegenerative diseases (ND) are indicated, while those of tumors (data were retrieved from COSMIC, ICGC, and TCGA databases) are shown in the Venn diagram. tRNA (black) binds to one lobe of Elp123 that is involved in Elp456 dynamic interactions. Elp3 hydrolyzes acetyl-CoA and cleaves SAM for the cm^5^ addition on U_34_. The cm^5^U_34_ is then converted to the derivatives by other enzymes. (**b**) Summary of reported alterations and PTM sites of mouse Elp3 (p-Ser161) and human Elp3 (p-Tyr202 and m-Lys229). The residue numbers and domains of human Elp3 are indicated. The locations of the alterations are as listed in the cartoon representation and they are color-coded according to which database resources are found. The experimentally verified residues are indicated with an asterisk. The identified PTM sites (solid circles) of Elp3 as well as the catalytic residues for acetyl-CoA binding/hydrolysis (open square) in Elp3 are listed.

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
