# Peer review of "How Elongator Acetylates tRNA Bases"

_ijms, 2020, doi:10.3390/ijms21218209_

Round 1
Reviewer 1 Report
The review of Abbassi et al. about the catalytic subunit of the elongator complex, Elp3, is informative and interesting and deserves publication. Elp3 is homologous to acetyl transferases that transfer the acetyl residue from acetyl-CoA to the N-terminus or to lysine residues of proteins. Yet, its substrate is not a protein or peptide but it transfers the acetyl group of Acetyl-CoA to N5 of a uridine residue at position 34 (the wobble position) of certain tRNAs.
The only serious point of critique is that, in the last section of the introduction, the authors should stress out more clearly that, in contrast to common (protein) acetyl transferases or NAT10, Elp3 does not simply catalyse the formation of a carbamoyl group. Rather, it connects the methyl group of the acetyl residue with the C5 atom of the uracil base, which requires the participation of an SAM radical enzyme domain. In addition, it should be mentioned if such a reaction is catalyzed by any other aceteyl transferase of the GNAT superfamily or if it is unique to Elp3.
Typos:
Line 38: change “sidechain” to “side chain”.
Line 119: change “archaea” to “archaeon” (archaea is plural).
Line 206: remove the comma after “reaction”.
Author Response
The review of Abbassi et al. about the catalytic subunit of the elongator complex, Elp3, is informative and interesting and deserves publication. Elp3 is homologous to acetyl transferases that transfer the acetyl residue from acetyl-CoA to the N-terminus or to lysine residues of proteins. Yet, its substrate is not a protein or peptide but it transfers the acetyl group of Acetyl-CoA to C5 of a uridine residue at position 34 (the wobble position) of certain tRNAs. The only serious point of critique is that, in the last section of the introduction, the authors should stress out more clearly that, in contrast to common (protein) acetyl transferases or NAT10, Elp3 does not simply catalyze the formation of a carbamoyl group. Rather, it connects the methyl group of the acetyl residue with the C5 atom of the uracil base, which requires the participation of an SAM radical enzyme domain. In addition, it should be mentioned if such a reaction is catalyzed by any other acetyl transferase of the GNAT superfamily or if it is unique to Elp3.
Response: Following the reviewers comment, we tried to emphasize the statement and have added two sentences in the last section of the introduction on page 2 (lines 78-81), which now read as follows “In contrast to the formation of a carbamoyl group by common protein acetyltransferases, Elp3 connects the methyl group of the acetyl residue with the C5 atom of the targeted uracil base. This type of reaction is unique to Elp3 proteins and in addition to a KAT domain, it also requires an additional rSAM domain to execute its full modification activity.”
Typos:
Line 38: change “sidechain” to “side chain”.
Line 119: change “archaea” to “archaeon” (archaea is plural).
Line 206: remove the comma after “reaction”.
Response: The typos have been corrected in the revised version.
Reviewer 2 Report
Acetylation occurs in a myriad of proteins as a posttranslational modification, and also to RNA molecules, all catalyzed by specific enzyme families. This review is focused on the catalytic subunit of Elongator complex, Elp3, which had been erroneously associated with protein acetylation and then turned out to be involved in tRNA-base acetylation. The modification of the tRNA antiocodon first base into xcm5U by Elongator is one of the most interesting topics in recent five years in this field, and there are indications that this also happens to Archaea. Whereas the modification of U34 into xnm5U in bacteria undergoes a distinct pathway, Elp3 is found in a particular bacterial species (Dmc), although the occurrence of xcm5U is not confirmed yet.
In this manuscript, the authors describe and compare the structures of three Elp3s from all of the different kingdoms, and then discuss the mechanism of catalysis. The requirement of tRNA for acetyl-CoA hydrolysis is surely the most fascinating in this mechanism, and waits for future studied to be fully understood. Recent studies showing the link between the dysfunction of Elp3 and neurodegenerative diseases and cancers are also reviewed, and finally, the authors mention that posttranslational modifications, like phosphorylation, can control Elongator function.
The tRNA modification by Elongator is an emerging research field possibly related to the causes of human illnesses. This mini-review is focused on Elp3 and successfully provides a concise view of the biological significance of Elogators. This manuscript is thus worth publishing as it is.
Author Response
Acetylation occurs in a myriad of proteins as a posttranslational modification, and also to RNA molecules, all catalyzed by specific enzyme families. This review is focused on the catalytic subunit of Elongator complex, Elp3, which had been erroneously associated with protein acetylation and then turned out to be involved in tRNA-base acetylation. The modification of the tRNA antiocodon first base into xcm5U by Elongator is one of the most interesting topics in recent five years in this field, and there are indications that this also happens to Archaea. Whereas the modification of U34 into xnm5U in bacteria undergoes a distinct pathway, Elp3 is found in a particular bacterial species (Dmc), although the occurrence of xcm5U is not confirmed yet.
In this manuscript, the authors describe and compare the structures of three Elp3s from all of the different kingdoms, and then discuss the mechanism of catalysis. The requirement of tRNA for acetyl-CoA hydrolysis is surely the most fascinating in this mechanism, and waits for future studied to be fully understood. Recent studies showing the link between the dysfunction of Elp3 and neurodegenerative diseases and cancers are also reviewed, and finally, the authors mention that posttranslational modifications, like phosphorylation, can control Elongator function.
The tRNA modification by Elongator is an emerging research field possibly related to the causes of human illnesses. This mini-review is focused on Elp3 and successfully provides a concise view of the biological significance of Elongator. This manuscript is thus worth publishing as it is.
Response: We highly appreciate the very positive feedback on our work.